# Retinal Ciliopathies and Potential Gene Therapies: A Focus on Human iPSC-Derived Organoid Models

**DOI:** 10.3390/ijms25052887

**Published:** 2024-03-01

**Authors:** Andrew McDonald, Jan Wijnholds

**Affiliations:** 1Department of Ophthalmology, Leiden University Medical Center (LUMC), 2333 ZC Leiden, The Netherlands; a.mcdonald@lumc.nl; 2Netherlands Institute of Neuroscience, Royal Netherlands Academy of Arts and Sciences (KNAW), 1105 BA Amsterdam, The Netherlands

**Keywords:** retinal ciliopathy, retinal organoid, cilium, gene therapy, CRISPR/Cas9, adeno-associated virus (AAV), *CEP290*, *USH2A*, *RPGR*, *MYO7A*

## Abstract

The human photoreceptor function is dependent on a highly specialised cilium. Perturbation of cilial function can often lead to death of the photoreceptor and loss of vision. Retinal ciliopathies are a genetically diverse range of inherited retinal disorders affecting aspects of the photoreceptor cilium. Despite advances in the understanding of retinal ciliopathies utilising animal disease models, they can often lack the ability to accurately mimic the observed patient phenotype, possibly due to structural and functional deviations from the human retina. Human-induced pluripotent stem cells (hiPSCs) can be utilised to generate an alternative disease model, the 3D retinal organoid, which contains all major retinal cell types including photoreceptors complete with cilial structures. These retinal organoids facilitate the study of disease mechanisms and potential therapies in a human-derived system. Three-dimensional retinal organoids are still a developing technology, and despite impressive progress, several limitations remain. This review will discuss the state of hiPSC-derived retinal organoid technology for accurately modelling prominent retinal ciliopathies related to genes, including *RPGR*, *CEP290*, *MYO7A*, and *USH2A*. Additionally, we will discuss the development of novel gene therapy approaches targeting retinal ciliopathies, including the delivery of large genes and gene-editing techniques.

## 1. Introduction

The retina is the tissue responsible for visual perception and is composed of multiple distinct cell types. Photoreceptors are the primary light-sensing neurons within the retina, responsible for photon capture and the initiation of the phototransduction cascade, which converts light stimuli into electrical-chemical signalling to be transmitted to the brain for interpretation. Both cone and rod photoreceptors are highly specialised cells containing stacks of membranous discs with photosensitive opsin pigments. These discs are located within a compartment known as the outer segment; in rods, each disc is surrounded by a separate plasma membrane and, in cones, all of the disc membranes are continuous with the outer segment plasma membranes (Figure 1). The outer segment lacks the capacity for protein synthesis and relies on the transport of proteins from the inner segment through the connecting cilium, a specialised microtubular structure essential for the maintenance and regulation of the outer segment (Figure 1) [1,2]. Impairment of the cilium leads to photoreceptor dysfunction and eventual cell death, which can be observed in the group of inherited retinal dystrophies that affect aspects of the cilium in the photoreceptors, broadly known as retinal ciliopathies [3,4].

Retinal ciliopathies can be caused by mutations in numerous ciliary genes affecting different aspects of cilial function, from cilial biogenesis to protein trafficking. Retinal ciliopathies can manifest as numerous diseases including retinitis pigmentosa (RP), Leber congenital amaurosis (LCA), cone-rod dystrophy, and macular degeneration, or as a component of several syndromic ciliary disease [5,6]. The diseases have diverse clinical presentations, ranging from early onset vision loss in childhood to slow progressive retinal dystrophy in later stages of life [6,7]. Accurate disease modelling is critical for understanding the pathology of diverse ciliopathies, and previous studies have primarily relied on rodents. Small or large animal models have several advantages since the structure and function of the retina can be analysed by non-invasive imaging such as scanning laser ophthalmoscopy, fluorescence angiography, spectral-domain optical coherence tomography, scotopic and photopic electroretinography, visual evoked potential testing, and optomotor and optokinetic response testing, and candidate drugs can be applied subretinal and intravitreal to the diseased retina containing blood vessels, microglial cells, and retinal pigment epithelium cells. However, rodent retinal disease models have limitations because of the differences in their photoreceptor structures and functions compared to humans [8]. Furthermore, adeno-associated viral (AAV) gene therapy vectors developed on animal disease models are sometimes not suitable for clinical studies. The AAV serotype but also the promoter, cDNA, or regulatory elements need optimization for human target cells in retinal explants or organoids.

The development of human-induced pluripotent stem cell (iPSC) technology offers a promising alternative. After informed patient consent and according to General Data Protection Regulation, patient somatic cells can now be harvested and reprogrammed into iPSCs, which can then be differentiated into a wide variety of cells and tissues. The iPSC-derived 3D retinal organoid, in particular, faithfully recapitulates retinal tissue (complete with Müller glial cells and all neuronal types), including rod and cone photoreceptors containing cilia with typical morphology and composition [9,10]. This presents an opportunity to study biomolecular function and structure in a human context, facilitating drug screening [11], and the development of gene therapies such as virally delivered gene augmentation and gene-editing strategies [12]. This review will discuss the state of human iPSC-derived technology in modelling retinal ciliopathies, including the limitations of current techniques, and the state of gene therapy-based approaches.

## 2. iPSC-Derived Retinal Organoids as Models for Human Retinal Ciliopathies

Accurate models of retinal diseases are crucial for understanding the underlying pathologies of inherited retinal disorders. Animal models have been extensively utilised to gain insights into these mechanisms, but while many aspects of retinal function are conserved across species, not all human genotype–phenotype relationships are faithfully replicated in model organisms. Inaccurate phenotypic manifestations hinder the examination of the underlying pathology leading to clinical presentations in patients.

Moreover, the broad heterogeneity of disease phenotypes observed in patients harbouring mutations in the same gene highlights the necessity for disease models that accurately display the relevant phenotype and share the same genetic background as the patient. Patient-derived iPSC models enable study in a precise genetic context, which may shed light on the observed heterogeneity. One of the most important stem cell technologies used for studying the human retina is the 3D retinal organoid. In 2011, Eiraku and colleagues pioneered the generation of 3D retinal organoids using murine embryonic stem cells [13]. It was discovered that embryonic stem cells could spontaneously and autonomously organise into stratified retinal tissue. Since then, other researchers have developed this concept and described methods for differentiating human iPSCs into 3D retinal organoids [9,10], which contain all major retinal cell types organised in a structure that closely resembles that of the human retina but lacks microglial cells and blood vessels in the retinal organoids and lacks a monolayer of retinal pigment epithelium aligned to the photoreceptors’ outer discs, the Bruch’s membrane, and the choroid. 

Bulk RNA analysis has revealed that human retinal organoids closely mimic the transcriptome of the human foetal retina [14,15]. Furthermore, single-cell RNA sequencing has confirmed this on a cell-to-cell basis [16,17]. The photoreceptor cilium structure in retinal organoids has been extensively examined using transmission electron microscopy (TEM), revealing a remarkable level of detail [18,19,20,21,22]. Furthermore, multiple studies have demonstrated the expression of numerous proteins associated with the cilium [23,24,25], suggesting a robust presence of functional cilial structure within iPSC-derived retinal organoids. However, it is worth noting that the photoreceptors of retinal organoids lack maturity and more closely resemble those of the foetal retina, with only rudimentary outer segments [10,20]. Nevertheless, photoreceptors in human iPSC-derived retinal organoids contain disc-like structures containing opsins and are capable of displaying light sensitivity [26,27]. 

Most retinal organoid differentiation techniques are highly similar, utilising largely similar medium components, with the most notable variation in the addition of particular small molecular compounds [28]. However, it is unclear which methodology can be defined as optimal as efficient differentiation appears to be largely iPS-cell-line-specific. Notably, the inclusion of foetal bovine serum (FBS) in a culture medium, a frequently used component, can inhibit ciliogenesis. FBS contains lysophosphatidic acid, which has been shown to interfere with the Akt signalling pathway, leading to a disturbance of ciliogenesis [29]. The use of serum replacement supplements has been shown to significantly elongate cilia and improve inner/outer segment thickness by differentiation day 200 [24].

Currently, the lack of fully developed outer segments is a major limitation of retinal organoids. This may be alleviated by continued fine-tuning of the culture conditions, for example, recent research has shown that antioxidant and lipid supplementation can improve outer segment structure [22]. However, perhaps the 3D retinal organoid model has limitations that cannot be circumvented by modifying the culture medium. Traditional methods of differentiation lack the spatial organisation of retinal pigmented epithelium (RPE) and the neural retina. Outer segment discs are regularly shed and phagocytosed by the RPE, and the interaction between photoreceptors and RPE may be critical in accurately mimicking the human retina [30]. Retina-on-a-chip is an emerging technology that aims to address this shortcoming by combining RPE cultures with retinal organoid to more accurately simulate the retinal microenvironment [31]. The lack of mature photoreceptor outer segments in retinal organoids may limit their viability as a model for inherited retinal ciliopathies. This is because the primary function of the cilium is to maintain the outer segment membranes and content. As a result, certain mutations may not recapitulate the observed phenotype in patients. Currently, major limitations of retinal organoids include cost, reproducibility, scalability, labour requirements, and long culture times. Human iPSC-derived retinal organoids mature relatively slowly, taking up to 8 months to reach maturity and requiring a prolonged use of often-expensive culture reagents. Moreover, regardless of the differentiation technique used, the production of retinal organoids requires the manual sorting of neural epithelium from undesired differentiated material, which is a time-consuming and laborious process. 

Inconsistencies in the ability to successfully differentiate certain iPSC lines, as well as the heterogeneity of organoids differentiated from the same iPSC line, still persist [32]. Variations in the differentiation capacity of particular iPSC lines may be due to lack of improper characterisations, as a number of common genetic aberrations can occur during reprogramming and subsequent culturing; therefore, it is vital to regularly perform analyses to confirm the genetic integrity of human iPSC lines [33,34,35]. These issues provide a significant obstacle to overcome in order to use retinal organoids as a truly consistent high-throughput model.

Various methodologies can be employed to assess the validity of a retinal organoid disease model and the efficacy of potential therapeutic interventions [36,37]. Primarily, the evaluation of retinal organoids is conducted on a morphological basis. Immunohistochemical analysis is commonly employed to validate the loss or mislocalisation of specific proteins of interest. There are well-defined expression profiles of particular retinal cells; these proteins are usually considered essential to the viability or function of the cell in question. In retinal organoid disease models, these specific cell markers may be mislocalised, reduced, or lost entirely, suggesting a perturbation of normal cellular function. A distinctive morphological feature of retinal degeneration involves the thinning of retinal layers following the death of particular cell populations. As retinal organoids are well-laminated with defined outer and inner nuclear layers, monitoring changes in retinal thickness over time serves as a valuable metric for assessing retinal degeneration [38,39]. Additionally, the transcriptome of retinal organoid disease models can be investigated, with alterations in differentially expressed genes offering insights into changes in specific retinal functions. The emergence of single-cell RNA sequencing provides high-resolution analysis, shedding light on cell-specific mechanisms [38,39]. Whilst animal retinal disease models permit various functional assays, retinal organoids are currently limited in this regard. The primary function of the retina, phototransduction, involves the detection of light stimuli and subsequent conversion into electro-chemical signalling. Techniques utilising patch clamps and microelectrode arrays can be employed to assess phototransduction in retinal organoids. Despite exhibiting demonstrable light responses, indicative of functional retinal neurons, retinal organoids have displayed limited phototransduction capacity [26,27,40]. In summary, the analysis of retinal organoids is primarily confined to morphological assessments as well as transcriptome and protein expression profiling, with a limited number of functional assays. It is arguable that retinal organoids possess certain limitations in terms of retinal analysis when compared to animal models. Consequently, retinal organoids currently serve as a valuable supplementation, rather than an outright replacement for animal models. Researchers should be aware of the distinct benefits and limitations associated with particular models.

Despite these challenges, the value of human-derived retinal organoids cannot be understated. They can replicate human retinal tissue to a relatively high degree of accuracy in terms of cell type, tissue structure, transcriptome, and cellular function. Many research groups have shown that retinal organoids faithfully recapitulate disease phenotypes and offer an invaluable resource for the direct study of underlying pathological mechanisms of retinal ciliopathies (Table 1) and other inherited retinal disorders in the human retina [18,23,38,41,42,43,44].

## 3. Retinal Organoid Ciliopathy Models

### 3.1. RPGR

Mutations in *RPGR* account for the majority (70–90%) of cases of X-linked RP (XLRP) and 10–20% of all incidences of RP [56]. Although the precise function of RPGR is not fully understood, studies indicate its involvement in protein trafficking at the transition zone of the photoreceptor cilium [57,58], as well as actin regulation and ciliogenesis [59,60]. RPGR exists in two primary isoforms: the constitutively expressed 1–19 isoform (RPGR^1–19^) and the photoreceptor-specific ORF15 isoform (RPGR^ORF15^). In the photoreceptor, RPGR undergoes differential splicing, resulting in the inclusion of an alternative 15th exon with its own stop codon and polyadenylation sequence [61]. This alternative open reading frame, known as ORF15, comprises a highly repetitive region encoding a domain rich in glutamic acid-glycine (Glu-Gly) amino acid residues. Due to its repetitive nature, the terminal exon is prone to mutations, with the majority of reported *RPGR* mutations occurring in this region [61,62], although mutations throughout the entire *RPGR^ORF15^* sequence can cause retinal disease [63,64]. Post-translational modification of the RPGR^ORF15^ isoform occurs through glutamylation within the Glu-Gly motifs of the ORF15 region. It appears that glutamylation is necessary for functional retinal RPGR, as the loss of glutamylation due to the mutation or ablation of the glutamylating enzyme TTLL5 results in a retinal phenotype [65]. 

RPGR XLRP has been successfully modelled using patient-derived retinal organoids with mutations in the ORF15 region [22,23,45]. In addition, an intronic variant causing abnormal splicing of intron 11 has also been studied [25]. These models have revealed consistent findings, including the mislocalisation of opsins, resembling the phenotype observed in animal models [66,67,68]. In these retinal organoids, a significant portion of rhodopsin and L/M opsin fails to be properly transported to the photoreceptor outer segments and remains localised in the inner retina. Furthermore, irregular rod morphology and reduced phototransduction have been observed [23]. The expression of RPGR protein itself is dramatically reduced [22], or exhibits abnormal localisation, failing to localise at the transition zone of the connecting cilium [25]. An apparent perturbation of ciliogenesis is present, as seen by significant decrease in cilial length in both retinal organoids and iPSC-derived RPEs.

*RPGR* has been implicated in the organisation of actin [59,60,69], and this effect has also been observed in *RPGR* mutant retinal organoid models, showing a significant increase in F-actin at the outer limiting membrane [25,45]. In a study by Megaw and colleagues, an elevated phosphorylation of cytoskeletal regulatory proteins was detected in *RPGR* mutant iPSC-derived photoreceptors, indicating a broad disruption in cytoskeletal regulation [45]. Specifically, it was found that *RPGR* mutations hindered gelsolin activation and subsequent downstream actions. However, it is worth noting that West et al. did not observe any irregularities in F-actin, suggesting that actin dysregulation may not be the sole mechanism underlying *RPGR*-mediated XLRP [22]. RPGR^ORF15^ has been identified as a primary substrate for glutamylation mediated by the enzyme TTLL5 [70]. Moreover, mutations affecting domains crucial for glutamylation or the elimination of TTLL5 function have been shown to replicate a retinal phenotype resembling *RPGR*-related conditions [65,70]. Studies conducted on *RPGR* knockout animal models have demonstrated a loss of glutamylation [65,71]. However, the current published research has not explored the glutamylation status of RPGR in iPSC-derived retinal organoids. The precise mechanism by which the absence of glutamylation contributes to the retinal phenotype remains unclear. It is possible that the lack of glutamylation leads to a disruption of protein–protein interactions within the protein complexes at the transition zone of the photoreceptors’ cilium. Whether this directly affects cytoskeletal regulation by interfering with interactions involving RPGR and gelsolin, or if it constitutes a distinct disease mechanism, requires further investigation. The absence of glutamylation may give rise to a unique pathological condition, such as mutations resulting in distal truncations in the C-terminal domain, which is essential for TTLL5 interactions and leads to a cone-dominant phenotype—in contrast to the N-terminal mutations associated with a rod-dominant phenotype [64].

### 3.2. CEP290

CEP290 (Centrosomal Protein 290) is a broadly expressed protein first described as a component of the centrosome [72]. CEP290 is found in the photoreceptor where it is localised in the transition zone of the photoreceptor cilium; several studies have shown CEP290 to have a critical role in ciliogenesis and ciliary trafficking [73,74,75]. Mutations in *CEP290* are associated with several syndromic ciliopathies, such as Joubert syndrome, Bardet–Biedl syndrome, nephronophthisis, Meckel–Gruber syndrome, and Senior–Loken syndrome [76]. These syndromic ciliopathies affect a wide variety of organs, resulting in a number of severe phenotypes in addition to retinal degeneration. Notably, some mutations of *CEP290* result solely in Leber congenital amaurosis type 10. This LCA phenotype is frequently caused by a c.2991+1665A>G mutation, with most patients carrying this variant in at least one allele [77]. This deep-intronic mutation generates an aberrant splice site resulting in the inclusion of a cryptic exon containing a premature stop codon (p.C998X) [78,79]. Crucially, this variant results in a small fraction of correctly spliced mRNA, subsequently producing residual full-length protein [78]. It is hypothesised that this low level of residual wild-type protein is sufficient for normal cilial function in most organs, thereby not resulting in the aforementioned syndromic phenotypes; however, LCA can still manifest as the highly specialised photoreceptor cilium is particularly sensitive to perturbations in cilial function due to the reduced levels of CEP290 protein. A clinical trial investigated the effectiveness of antisense oligonucleotide-based therapy, which can inhibit the aberrant splicing of the cryptic exon; following an intravitreal injection, a transient visual improvement was recorded up to 4 years post-injection [80]. 

Earlier attempts to produce an animal model that accurately mimicked this cryptic splicing pathology were unsuccessful; a humanised mouse model carrying the specific c.2991+1655A>G intronic mutation failed to exhibit a retinal phenotype, likely due to limited recognition of the cryptic splice site by the endogenous mouse spliceosome [81,82]. Therefore, a fully human-derived disease model was required for recapitulation of the splicing events, as seen in patients with LCA10. With this aim, patient-derived iPSCs carrying homozygous c.2991+1665A>G mutations were first successfully differentiated into retinal organoids in 2016 [18]. The generated retinal organoids expressed both wild-type and aberrantly spliced mRNA, with the wild-type transcript accounting for as little as 10% of the total transcript pool. Although CEP290 protein was not detectable at the connecting cilium in mutant organoids, the organoids still developed in a typical manner, expressing photoreceptor markers such as recoverin and various opsins at the expected time points. The primary phenotype was observed in the significant reduction in ciliated cells and the reduction of cilial length. These results were corroborated in the following studies, in which patient-derived retinal organoids carrying a homozygous c.2991+1665A>G mutation were also generated [49,50]. In addition to the frequent aberrant splice variant, organoids derived from patients with LCA harbouring a compound heterozygous mutation (c.2991+1655A>G p.C998X + c.5668G>T p.G1890X) were also investigated [47,48]. Findings from these organoids generally aligned with the results of homozygous c.2991+1655A>G, albeit with a more severe defect in rod development. Analysis of these organoids with transmission electron microscopy highlighted mother centrioles docked to the cell membrane without further cilial development, suggesting a defect in ciliogenesis in patient-derived organoids.

The bulk of current organoid research regarding the retinal *CEP290* phenotype has understandably been focused on the c.2991+1655A>G variant, but the retinal phenotype arises in many other *CEP290*-related syndromic diseases. In relation to this, a recent study made use of CRISPR/Cas9 technology to generate a compound heterozygous *CEP290*-null organoid model [50]. The modification in exon 6 lead to single base pair deletions (c.C315del + c.T316del; p.Ser104Leufs*20). The authors suggest the edited line should resemble a pathogenic variant (c.338T>A, p.Leu113*) found in Joubert and Meckel–Gruber syndromes. The *CEP290*-null organoids displayed a reduction in cilia number and length in a similar manner to the LCA patient-derived organoids. Without further study, it is unclear if *CEP290*-null organoids would display any other phenotypic variances. It is possible that as the CEP290 protein level is already below the pathogenic threshold, the complete depletion of the protein may not significantly exacerbate the retinal phenotype.

### 3.3. Usher Syndrome

Usher syndrome is an autosomal recessively inherited disorder classically presenting visual, auditory, and vestibular impairment, and is divided into three distinct subtypes classified by time of onset and severity of phenotype, with type 1 being the most severe [83,84]. Usher syndrome is heterogenous, both in terms of clinical presentation and genetic origin, with a number of diverse genes known to carry pathogenic mutations (Type 1: *MYO7A*, *USH1C*, *PCDH15*, *CDH23*, *USH1G*, and *CIB2*; Type 2: *USH2A*, *ADGRV1*, and *WHRN*; Type 3: *CLRN1*) [85,86]. The proteins derived from these genes vary in structure and function but have been shown to primarily localise at the periciliary region or calyceal processes of the photoreceptor, playing a critical role in maintenance of outer segments [87,88]. There is a body of evidence for the direct interaction between many Usher proteins in a form of interactome, hence why a disruption in different genes leads to similar clinical presentation [88,89,90]. Attempts have been made to produce suitable animal models but there was often a lack of clear retinal phenotype similar to that found in humans, possibly due to variations in the protein expression profile, as well as the structural variations in photoreceptors between species [91,92]. With the hopes of more accurate models, some groups have made use of iPSC-derived models to better understand human disease mechanisms. So far, mutations in two Usher-syndrome-associated genes have been investigated in human iPSC-derived retinal organoids.

#### 3.3.1. *MYO7A*

Usher syndrome type 1 (USH1) represents the most severe form of the disease and is associated with mutations in six different genes. Amongst these, USH1B is characterised by mutations in the *MYO7A* gene, which is the most common mutation found in USH1, accounting for approximately half of all cases [83,93]. *MYO7A* encodes for myosin VIIA, a member of the myosin superfamily of molecular motor proteins [94]. Myosin VIIA and other USH1 proteins are essential for the structural integrity of photoreceptors by coupling the photoreceptor outer segment and the calyceal processes, the actin-filled protrusions of the inner segments which encompass the base of the outer segment [87,90]. Unfortunately, there are currently no suitable animal models available for studying USH1B. Mice lacking myosin VIIA (*Shaker1* mice) exhibit auditory deficiencies but fail to recapitulate the retinal phenotype seen in human USH1B. This discrepancy is likely due to the absence of calyceal processes in the mouse retina, which are essential for understanding the retinal manifestations of USH1B [87,91,92].

To date, only one published study has explored the generation of patient-derived retinal organoids harbouring mutations in *MYO7A* [55]. In this study, organoids were generated from iPSCs obtained from three individuals: two siblings carrying a heterozygous c.6070C>T (p.Arg2024*) + c.223G>C (p.Asp75His) mutation and a third patient with a compound heterozygous genotype carrying one allele with c.1996C>T (p.Arg666*) and the other allele with c.133-2A>G, resulting in non-canonical splicing in exon 4. Surprisingly, retinal organoids generated from these patients showed little morphological difference to controls, expressing typical late-stage photoreceptor proteins, such as ARR3 and rhodopsin. Additionally, no evidence of retinal degeneration was observed, and no elevated stress markers were detected with immunohistochemistry. 

To analyse the cell-specific effects of *MYO7A* mutations, the authors performed single-cell RNA sequencing of late-stage retinal organoids. *MYO7A* was found to be expressed in the Müller glial, bipolar, rod, and cone cells. Notably, Müller glial cells also expressed the other USH1-associated genes *USH1G*, *USH1C*, *CDH23*, and *CIB2*, suggesting that Müller glial cells may be more mechanistically relevant to Usher syndrome pathology than previously considered. Investigation into the differentially expressed genes revealed rod-specific upregulation in cellular responses to oxidative stress and pro-apoptotic genes. Of all *MYO7A*-expressing cells, Müller glial cells were found to be most numerous, and differentially expressed genes highlighted an increase in apoptotic-related pathways; this is interesting as Müller glial cell death is not considered a primary factor in USH1 retinal degeneration [83,95]. Despite elevated *MYO7A* transcription, localisation of myosin VIIA protein was not observed in Müller glial cells with immunohistochemistry. Although still unclear, the findings from this study hint at a mechanistically relevant role for Müller glial cells in Usher syndrome that warrants further investigation. 

More typical clinical manifestations of USH1B, such as degeneration of the photoreceptor layer was not observed; this could be explained in part by the immaturity of photoreceptors in retinal organoids. The bulk transcriptome profile of these organoids was found to more closely resemble that of the foetal retina rather than the adult one, which is in line with the findings of others [17,96]. Retinal organoids have been described to produce only rudimentary outer segment-like structures [10,20]; this could result in a lack of the calyceal process/outer segment interface thought to be an integral function of myosin VIIA. Such structural deviations from the adult human retina highlight a possible limitation of USH1B retinal organoid models.

#### 3.3.2. *USH2A*

Mutations in *USH2A* are the most commonly observed among all Usher-syndrome-related genes [93] and are also frequently found in autosomal recessive non-syndromic retinitis pigmentosa [97]. The *USH2A* gene encodes Usherin, a large transmembrane protein consisting of 5202 amino acids [98]. Usherin is localised at the periciliary membrane, forming a complex with other USH2 proteins, ADGRV1 and Whirlin. This complex directly links the periciliary membrane to the connecting cilium [88]. Usherin has also been implicated in vesicle trafficking toward the outer segments of photoreceptor cells [99]. Pathogenic mutations in *USH2A* have been identified throughout the entire gene, but the most frequent mutation is c.2299delG (p.Glu767fs) in exon 13, accounting for approximately 25% of all *USH2A* pathological variants [100,101].

Murine models with *Ush2A*-null mutations exhibit a retinal phenotype, although it is relatively mild and late onset, suggesting that these models do not fully replicate the phenotypic characteristics of patients with USH2 [102]. In contrast, several robust zebrafish models with *USH2A* mutations have been developed, displaying a more clinically relevant retinal phenotype [103,104]. However, it is important to consider that zebrafish and humans have substantial anatomical and physiological differences in the retina due to their evolutionary distance.

Retinal organoids have been successfully generated from a patient with non-syndromic retinitis pigmentosa associated with a novel pathogenic mutation in *USH2A* (c.8559-2A>G + c.9127_9129delTCC). Analysis of these organoids reported a significant reduction in the thickness of the retinal neuroepithelium and disrupted photoreceptor development. Furthermore, there was a notable decrease in the expression of laminin and other basement membrane markers, suggesting abnormalities in the extracellular matrix [52]. Abnormal formation of the RPE was also observed, characterised by a loss of pigmentation and reduced expression of typical markers—MITF, ZO-1, and RP65. A subsequent study further investigated the organoids derived from the same patient line using transcriptomic and proteomic analysis [53]. The primary findings revealed an upregulation of apoptotic processes and a dysregulation of extracellular matrix organisation, as well as microtubule cytoskeleton organisation and transport. 

More recently, another group generated retinal organoids carrying mutations in *USH2A*. Retinal organoids were derived from either patients with Usher syndrome carrying the prominent homozygous c.2299delG or patients with non-syndromic RP harbouring compound heterozygous c.2299delG and c.2276G>T mutations [54]. Surprisingly, distinct retinal phenotypes emerged, depending on the origin of the patient-derived retinal organoids. In organoids generated from patients with RP, a mislocalisation of Usherin, and its interacting partner ADGRV1 were observed; however, both proteins appeared to be entirely absent in Usher syndrome patient-derived organoids. Additionally, organoids derived from patients with RP exhibited a notable reduction in the length of the brush border, indicative of a reduction of inner/outer segment formation. This was further observed with TEM analysis, which revealed a significant reduction in the length of the inner/outer segment-like structures. Further analysis of these organoids discovered a rod phenotype characterised by a significant reduction of rod-specific proteins rhodopsin and PDE6β. This phenotype was not observed in retinal organoids derived from patients with Usher syndrome. In contrast, organoids derived from patients with Usher syndrome displayed a cone-specific phenotype, with a pronounced increase in the expression of cone markers ARR3 and R/G opsin. The researchers further analysed this striking disparity in phenotypes by comparing these results with clinical data. Interestingly, it was discovered that, of the patients with *USH2A* mutations, there is an increased incidence of cone-specific macular degeneration in patients classified with Usher syndrome compared to patients with non-syndromic RP, reflecting the observed differential phenotype in the retinal organoid models. This study exemplifies the power of patient-derived retinal organoids for investigating mutation-specific phenotypic variations.

## 4. Retinal Gene Therapy

### 4.1. AAV-Mediated Gene Supplementation

The retina is a desirable target for virally delivered gene therapies as the healthy eye is an immune-privileged organ with a reduced risk of triggering a strong immune response. Additionally, the eye is accessible with minor invasive surgery, allowing for the administration of gene therapy vectors via intravitreal or subretinal injections. Moreover, the post-mitotic nature of the retina allows for a sustained expression of the delivered transgene, potentially providing long-term therapeutic benefits. In the context of inherited retinal diseases, many of which are monogenetic, the retina offers an opportunity to replace or supplement the expression of a single gene to correct the underlying phenotype.

The adeno-associated virus (AAV) is the most commonly utilised vector for virally delivered gene therapies due to its low immunogenicity and low risk of integration [105]. The first FDA-approved AAV-delivered gene therapy was for the treatment of Leber congenital amaurosis type 2, in which a copy of *RPE65* was delivered to the retinal pigment epithelium via AAV serotype 2. The success of this therapy has paved the way for other AAV-delivered retinal gene therapies currently undergoing clinical trials [106,107]. 

Despite many promising developments, AAV-mediated gene therapy has a notable drawback with a limited packaging capacity of ~4.8 kb. Whilst suitable for a number of ciliopathy-related genes, such as *RPGR* (3.5 kb), many genes are beyond this capacity, including *CEP290* (7.4 kb) and several of the Usher-syndrome-associated genes (*MYO7A*: 6.6 kb, *CDH23*: 10 kb, *ADGRV1*: 18.9 kb, *USH2A*: 15.6 kb). As a result, the delivery of such genes would rely on alternative vectors or multi-AAV delivery strategies. Efforts are being made to develop and optimise these alternative strategies to overcome the packaging limitations of AAV and enable us to administer effective gene therapies for a broader range of retinal diseases.

### 4.2. Retinal Organoids as Platforms for Developing Virally Delivered Gene Therapies

The development of new gene therapies requires rigorous testing and proof of concept, typically using animal models. However, there are potential issues with animal models faithfully recapitulating the human retinal system, in addition to ethical considerations. In this regard, iPSC-derived retinal organoids provide an excellent platform for the testing of potential gene therapies. As previously discussed, they contain all the major retinal cell types and retain many functional aspects of the human retina.

Multiple studies have detailed the tropism of AAV serotypes in retinal organoids, demonstrating their utility as platforms for AAV-delivered gene supplementation [38,39,41,108,109,110,111,112]. Several studies have also demonstrated the efficacy of AAV gene therapies in retinal organoids as they restore gene expression in the desired cell types and alleviate the observed phenotype [38,39,43,44,113]. Therefore, the use of human iPSC-derived retinal organoids as a testing platform for gene therapies offers a valuable alternative to animal models, providing a more accurate representation of the human retinal system and avoiding ethical concerns associated with animal experimentation.

### 4.3. Targeting of Gene Therapies to the Photoreceptor

AAV vectors exhibit diverse cellular tropism, and within the context of retinal ciliopathies, the ideal target cell for gene therapy vectors is the photoreceptor. AAV2 has commonly been used as a delivery vehicle in the retina and is the primary serotype employed in human clinical trials for retinal gene therapy [106]. Additionally, other serotypes have demonstrated the ability to transduce photoreceptors [114]. Amongst animal models, AAV8 and AAV5 have shown evidence to be some of the most effective at photoreceptor transduction [115,116,117], with transduction also being observed in human retinal explants [41,118,119].

In addition to naturally occurring AAVs, research has focused on developing novel capsids with modified tropism and enhanced transduction efficiency. The tropism of AAV2-7m8, a modified version of AAV2, has been evaluated in retinal organoids, demonstrating a significantly improved transduction of photoreceptors [108,120]. Modified AAV6 (ShH10), AAV8, and AAV5 have been identified as the most effective in targeting the photoreceptor layer of human-derived retinal organoids [41,110]. The efficiency of AAV2-7m8 and ShH10 in transducing photoreceptors of retinal organoids has been further confirmed in subsequent studies [121]. Similarly, modified AAV9 has also demonstrated an enhanced transduction efficiency, with two specific variants—AAV9.GL and AAV9.NN—outperforming AAV2 and unmodified AAV9 in retinal organoids [109]. K912, an AAV variant generated through directed evolution, outperformed a number of naturally occurring and modified AAV capsids in non-human primates and human retinal explants [119,122]. Continued advancements in novel capsid design hold great promise for the development of even more robust vectors specifically tailored for targeting photoreceptors upon intravitreal or subretinal injections. 

Whilst specific AAV capsids have shown a propensity for targeting photoreceptors, it is inevitable to encounter the transduction of undesired cell types. To achieve a truly photoreceptor-specific expression of a transgene, a combination of cell-specific promoters and efficient AAV capsid targeting is necessary. Two well-validated photoreceptor-specific promoters are the human interphotoreceptor retinoid-binding protein (IRBP) [123] and the rhodopsin kinase promoter (GRK1) [124,125]. Amongst them, the GRK1 promoter stands out as an optimal choice for AAV therapies targeting photoreceptors due to its strength, specificity, and compact size, allowing a significant portion of the viral carrying capacity to be dedicated to the transgene. The GRK1 promoter has been demonstrated to induce photoreceptor-specific expression, even when using broadly transducing serotypes in retinal organoids [111]. This promoter has been utilised in at least three clinical trials of AAV-mediated gene therapies targeting ciliary genes (NCT03116113, NCT03872479, NCT03316560). Additionally, a recent study employed a promoter derived from the human cone-rod homeobox protein (CRX) to achieve highly specific photoreceptor expression in an organoid model derived from a patient with Leber congenital amaurosis [44]. The strategic combination of cell-specific tropism and promoters plays a pivotal role in the development of an effective and precise gene therapy approach.

### 4.4. AAV-Mediated Rescue of Cilial Genes in Retinal Organoid Models

AAV-mediated gene delivery has been successfully employed in retinal ciliopathy organoid models. An LCA organoid model harbouring mutations in the ciliary gene *NPHP5* demonstrated effective rescue of a cilial protein using AAV2-delivered *NPHP5* under a ubiquitous CMV promoter [24]. The gene therapy resulted in notable increased thickness of the outer segment layer and corrected localisation of rhodopsin from the ONL to the outer segments, as well as restoring the localisation of CEP290 to the connecting cilium.

A *RP2*-null X-linked RP model derived from CRISPR/Cas9-edited iPSC lines has been used to assess AAV5-delivered *RP2* gene therapy [43]. The organoids were transduced at DD140, showing strong transduction of the outer nuclear layer, with a confirmed expression of *RP2* in both cones and rods. Despite the use of a general CAG promoter, the expression was primarily observed in the photoreceptor layer. A rescue of diseased organoids was achieved with not only a strong increase in RP2 expression but also a thickening of the outer nuclear layer, and an increase of rhodopsin expression 40 days post-transduction.

Ciliopathy-targeted gene therapy was also demonstrated in an XLRP model derived from patients with *RPGR* mutations. AAV2-7m8 was used to deliver an abbreviated form of *RPGR^ORF15^* under the control of the photoreceptor-specific GRK1 promoter [22]. The restoration of RPGR protein to the cilium was confirmed through immunohistochemistry, and mislocalised rhodopsin was redirected above the outer nuclear layer, suggesting rescue of functional cilial transport. This transgene had previously shown efficacy in a murine *RPGR* disease model [126] and has now been validated in retinal organoids, demonstrating the feasibility of using organoids as alternatives for animal models. Encouraging experimental evidence for *RPGR* AAV-mediated gene therapy has led to several human trials [107,127]. These retinal organoid studies, which demonstrate efficient delivery and measurable rescue, provide compelling evidence for the potential use of retinal organoids in testing AAV-mediated gene therapies for retinal ciliopathies in human patients.

## 5. Alternative Approaches for the Delivery of Large Cilial Genes

Many ciliopathy-related genes are too large to be delivered by a single AAV due to the relatively small packaging capacity of ~4.8 kb. Alternative strategies exist where two or more AAV vectors can be used in conjunction, which then recombine to produce a full-length sequence. The instance of recombination can be achieved at either the DNA, RNA, or protein level [128,129].

The process of combining viral genomes is known as AAV trans-splicing—where the expression construct is split in two, with one vector containing the promoter, the 5′end of the coding sequence and a splice donor; the second viral vector contains a splice acceptor, the 3′ end of the coding sequence, and a polyadenylation sequence. Following transduction, the viral genomes can combine either due to the inclusion of specific homologous sequences facilitating homologous recombination, or via the concatemerization mediated by the flanking inverted terminal repeat (ITR) sequences present in both vectors. Following transcription of the sequence on the concatemerized DNA, the ITR junction is spliced out, resulting in the full-length mRNA of the desired gene (Figure 2A) [130,131]. Dual trans-splicing vectors have been used to achieve expression of ABCA4 (6.8 kb) and myosin VIIA (6.6 kb) in both mice and pigs [132]. Furthermore, a triple trans-splicing system has been employed to deliver ~14 kb in animal retina, which permitted the delivery of the USH1D-associated gene *CDH23* (10 kb) [133]. It should be noted that the increased capacity offered by this system comes at a price, as there was a substantial drop in the expression efficiency of the triple AAV system when compared to a single AAV. This reduced efficiency is most likely due to the inherent complexity of the system which requires multiple sequential steps to achieve expression, including multiple AAV transduction, proper concatemerisation, mRNA splicing, and translation. 

A similar technique exists known as protein trans-splicing, which relies on split-inteins, which are genetic elements that allow for the recombination of protein sequences [134]. In this context, each vector would deliver a fragment of the protein of interest adjoined with either a N- or C-terminal intein. Following the recombination of the split-intein, the element subsequently removes itself resulting in unaltered full-length protein (Figure 2B). The recombination and protein splicing event happens spontaneously following translation and does not rely on any additional co-factors [135]. One consideration is that following excision, the inteins would persist in the eye and, due to not being of mammalian origin, could elicit an immune response in the host [136]. Protein trans-splicing was successfully implemented to reconstitute full-length CEP290 and ABCA4 in both animal and retinal organoid models, with intein-mediated recombination demonstrating a greater efficiency compared to AAV-genome-based trans-splicing [137]. The efficiency of reconstitution is highly dependent on the position of the split-intein, and careful consideration should be applied as not to negatively affect protein folding. Additionally, as the system requires both halves to be independently transcribed a promoter needs to be included in both cassettes, taking up valuable space within the AAV vectors’ limited cargo capacity [137]. 

Recently, the available toolkit for the delivery of large genes has expanded with a technique based on mRNA trans-splicing known as REVeRT (Reconstitution via mRNA trans-splicing) [129]. mRNA trans-splicing is achieved by including complimentary binding domains in both sequences intended to be reconstituted. Following transcription, the two independent mRNA sequences can bind via the complimentary sequences. The inclusion of splice sites flanking the binding domains facilitates the removal of these additional sequences, resulting in the production of a mature, full-length mRNA (Figure 2C). This innovative approach was successfully utilised for the expression of large Cas9 fusion proteins, including CRISPRa and prime editor. Furthermore, this technique was applied to reconstitute full-length ABCA4 in the retinas of mice. The authors found mRNA trans-splicing to be of comparable efficiency to split-intein systems, whilst offering several advantages. In mRNA trans-splicing, there is a greater degree of flexibility in choosing a split site, in contrast to split-intein approaches where appropriate selection of the split site is essential to maintain correct protein folding. Additionally, mRNA trans-splicing does not generate the excised intein, which is a potentially immunogenic by-product.

For extremely large genes, such a *USH2A*, a multiple vector system would be required to sufficiently encompass the entire coding sequence along with the promoter and other essential sequences. However, utilising multiple vectors results in a significant decrease in expression efficiency, making such strategies impractical for very large genes, such as *USH2A*. In these instances, alternative viral vectors may offer a solution. High-capacity adenoviral vectors (HC-AdV), with a packaging capacity of up to 36 kb, have been shown to effectively transduce retina cells (Figure 2D) [138,139,140,141]. Most research has focused on human adenovirus type 5 (HAdV5) and its modified variants, as the various subtypes of HAdV5 exhibit efficient transduction and diverse cellular tropism [142,143]. The 3rd generation of gutless adenoviral vectors are devoid of viral genes and have been demonstrated to elicit a reduced immune response [144]; however, concerns still persist around potential inflammatory responses following HC-AdV transduction [145,146]. Currently, there are no clinical trials utilising adenovirus as a vehicle for gene supplementation in the retina. Although high-capacity adenoviral vectors hold promise for large gene delivery, a further investigation is needed to characterise these vectors for use in the human retina.

## 6. Gene-Editing Approaches in the Retina 

Gene editing represents a promising therapeutic approach that offers a more enduring solution in comparison to gene supplementation. The widely used CRISPR/Cas9 system utilises programmable guide RNAs (gRNAs) to direct the Cas9 nuclease to the target location in the genome, inducing a double-strand break (DSB) [147]. This powerful mechanism can be harnessed to knock-out-specific genes or facilitate the insertion of a donor sequence at the DSB site utilising endogenous DNA repair pathways [148,149]. Recent advancements in this technology have given rise to novel editing systems like base and prime editors [150,151]. The field of gene editing is rapidly evolving, expanding the repertoire of available tools and enabling a wider range of edits with increasing precision. In a notable breakthrough, the first CRISPR/Cas9-mediated gene-editing therapy was successfully employed in the human retina to correct the deep intronic c.2991+1655A>G mutation of the ciliary gene *CEP290* (NCT03872479). An AAV5 vector was used to deliver *S. aureus* Cas9 and two gRNAs targeting regions flanking the intronic cryptic splice site. Cleavage at these positions resulted in the excision of the cryptic splice site, allowing for the expression of wild-type protein [152]. Whilst this milestone represents significant progress in human gene-editing therapies, it should be noted that this particular technique is applicable only to a small subset of inherited retinal diseases. Many mutations necessitate the correction or insertion of a new sequence rather than the deletion of a genomic segment.

### 6.1. Prime Editing

Prime editing is an innovative technique that shows promise for precisely repairing short regions of the genome [153]. It utilises a fusion of Cas9-nickase and reverse transcriptase to create prime editors. This system employs a modified guide RNA called a pegRNA, which not only directs the Cas9 to the target site but also serves as a template for the reverse transcriptase. The reverse transcriptase synthesises a newly corrected DNA strand, which is then incorporated into the genome at the site of the DNA nick. One of the main advantages of prime editing is its ability to avoid inducing double-strand breaks, thus reducing the risk of introducing small insertions or deletions (indels) and chromosomal rearrangements [150]. This strategy has been successfully employed to rescue a retinal phenotype in LCA mouse models by correcting mutations in *RPE65* [154,155]. However, it should be noted that this technique is currently limited in application due to the relatively small size of possible edits, typically insertions of around 44 bp. To achieve broad applicability, individual therapies would need to be developed for each specific pathogenic mutation. Nevertheless, it could have meaningful utility in treating mutations that occur at a high frequency, such as the c.2299delG mutation found in *USH2A*. Delivery of this system also presents a challenge as the entire prime editing cassette is approximately 6.5 kb, exceeding the cargo capacity of AAV. As discussed previously, dual trans-splicing systems offer a solution to delivering large cargos. Recently, the AAV-mediated delivery of split primer editor, reconstituted by a split-intein system and split-RNA system, has been demonstrated in the mouse retina with a high degree of efficiency [129,155,156]. In conclusion, prime editing is a versatile tool that can be applied to a number of pathogenic mutations, enabling safe and precise corrections. The integration of prime editors with large-capacity delivery systems holds promise for the future treatment of inherited retinal disorders.

### 6.2. Homology-Independent Targeted Integration

Ideally, a gene-editing strategy should be versatile enough to address a wide range of mutations within a specific gene. This approach would involve the insertion of the complete coding sequence of the gene, akin to AAV gene supplementation strategies, but utilising gene-editing techniques. The successful demonstration of DNA sequence insertion into the genome has been observed in various cases; typically, these knock-in strategies leverage the endogenous homology-directed repair (HDR) DNA repair pathway by introducing a donor sequence flanked by homology arms that precisely match the desired insertion site in the genome adjacent to the Cas9 cleavage site [157,158]. However, applying these strategies to the retina poses a challenge since the retina is a mostly non-dividing tissue, and HDR is not active in non-dividing cells [159]. In contrast, the other major DNA repair pathway, non-homologous end joining (NHEJ), is active throughout the cell cycle but is known to be error-prone, resulting in indels at a relatively high frequency [160]. The ideal gene-editing strategy would combine the precision of HDR with the constant activity of NHEJ.

A gene-editing technique, pioneered by Suzuki and colleagues in 2016 and known as homology-independent targeted integration (HITI), demonstrated the ability to successful insert sequences lacking homology in post-mitotic tissue with a high degree of precision and relatively low indel formation [161]. In this system, the donor sequence is flanked by the same gRNA target site as in the target genomic locus. Remarkably, it was discovered that cleaving the donor sequence in addition to the desired genomic locus can lead to significantly improved integration with few unwanted events. This strategy’s effectiveness was demonstrated in a retinitis pigmentosa rat model with a deletion in the *mertk* gene, and a significant phenotypic rescue was observed with MERTK protein expression in retinal pigment epithelium restored along with visual function. Moreover, the HITI technique was also recently used to knock-in the optogene Opto-mGluR6 into the ON-bipolar cells of rd1 mice, restoring light-sensitivity to the retina [162]. Another notable application of HITI was the treatment of an autosomal dominant *RHO* mutation. In instances such as this, it is required to ablate the disease-causing allele whilst also supplying a new healthy copy of the *RHO* gene. In an elegant system, this was achieved with a single editing event; in the endogenous *RHO* locus, a cassette was inserted beginning with a stop codon to block endogenous expression subsequently followed by a healthy copy of *RHO* complete with its own transcription start site and polyadenylation sequence. This strategy resulted in transgene expression and showed an improvement in the retinal phenotype of mice with *Rho*^P23H−/+^ [163]. HITI-based gene editing holds significant promise for efficient and accurate gene editing in the retina. However, as with any method involving double-strand breaks, safety concerns still need to be addressed. Nevertheless, the outlook is promising, and the field will evolve as more precise gene-editing techniques continue to emerge.

## 7. Concluding Remarks

In conclusion, hiPSC-derived retinal organoids prove to be a powerful tool for investigating retinal ciliopathies, recapitulating expected phenotypes associated with mutations in a number of cilial genes. The retinal organoid platform presents a unique opportunity to investigate disease pathologies in a patient-derived context, offering insights that may be lacking in alternative models. Additionally, hiPSC-derived retinal organoids emerge as an excellent resource for the development and validation of potential gene therapies targeting retinal ciliopathies. Several innovative techniques for the delivery of large cilial genes are in development, broadening the range of potentially treatable ciliopathies. Whilst retinal organoids recapitulate human retinal tissue in a remarkably faithful manner, a few differences remain, most notably regarding the lack mature outer segment formation. This may lead to inaccurate disease modelling in the context of certain mutations; therefore, researchers must carefully consider if retinal organoids are the optimal model for their investigations. 

## Figures and Tables

**Figure 1 ijms-25-02887-f001:**
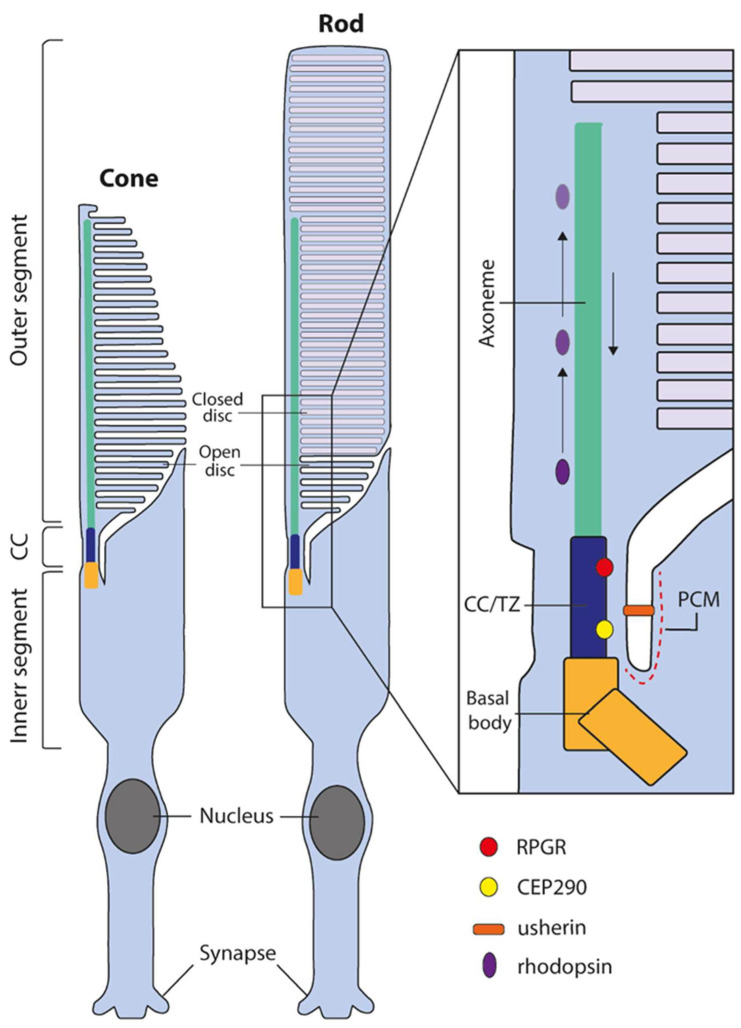
The structure of human cone and rod photoreceptors and connecting cilium. Both cone and rod photoreceptors are divided into two distinct sections, the inner segment, and the disc containing outer segment. The region connecting these distinct compartments is known as the connecting cilium or transition zone (CC/TZ). The outer segment lacks the capacity for protein synthesis and is dependent on protein trafficking across the CC/TZ and subsequent transport along the distal axoneme. Numerous proteins are required for cilial function and maintenance, such as RPGR and CEP290 localized in the CC/TZ or usherin located in the periciliary membrane (PCM). Black arrows depict anterograde and retrograde cilial transport.

**Figure 2 ijms-25-02887-f002:**
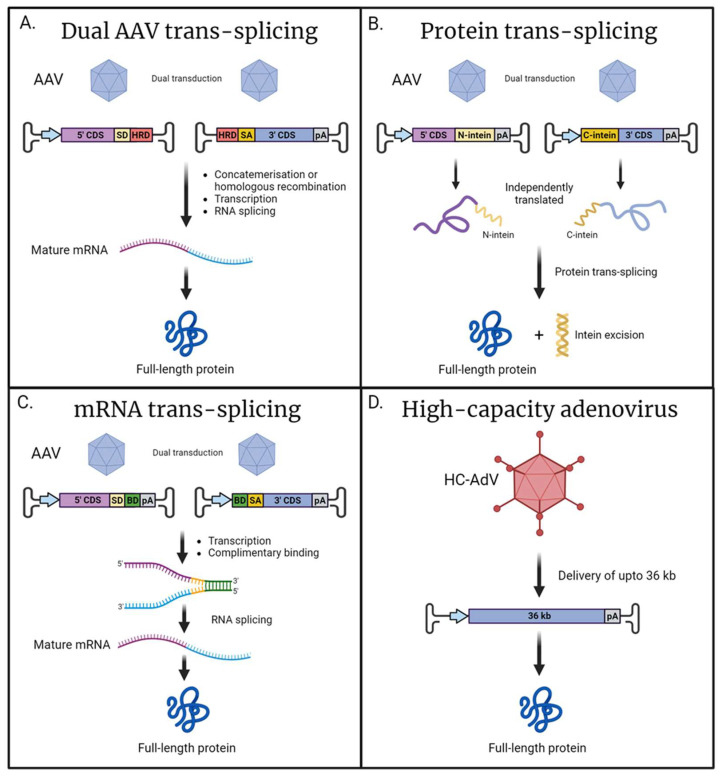
Alternative approaches for the delivery for large cilial genes. (**A**) Dual AAV trans-splicing: Two independently delivered viral genomes containing either the 5′ or 3′ coding sequence (CDS) recombined via concatemerization or homologous recombination mediated by the homology recombination domains (HRDs). The inclusion of splice donor (SD) and splice acceptor (SA) sequences permits RNA splicing to produce a mature mRNA encoding for full-length protein. (**B**) Protein trans-splicing: The CDS is split across two independent viral vectors resulting in the expression of either N- or C-terminal polypeptides of the desired full-length protein. Adjoined to these peptides are either the N- or C-terminal of a split-intein; post-translationally, the split-intein spontaneously recombines and self-excises, resulting in full-length protein. (**C**) mRNA trans-splicing: dual AAVs result in the transcription of two independent mRNAs encoding each half of the desired sequence, and the inclusion of complementary binding domains (BDs) allows for recombination. Flanking SD and SA sites facilitate RNA splicing and the removal of the BD sequences, resulting in a mature, full-length mRNA. (**D**) High-capacity adenovirus has a large cargo capacity of up to 36 kb; this facilitates the delivery of large genes in their entirety with a single vector. Promoter (blue arrow); polyadenylation signal (pA) (created with BioRender.com).

**Table 1 ijms-25-02887-t001:** Human iPSC-derived retinal organoid models of retinal ciliopathies.

Associated Disorder	Gene	Mutation	Reference
X-linked retinitis pigmentosa	*RPGR*	c.ORF15+689−692del4	[45]
X-linked retinitis pigmentosa	*RPGR*	c.1685_1686delAT c.2234_2235delGA c.2403_2404delAG	[23]
X-linked retinitis pigmentosa	*RPGR*	c.ORF15+652_653delAG c.ORF15+594_595delGA	[22]
X-linked retinitis pigmentosa	*RPGR*	c.1415 − 9A>G	[25]
X-linked retinitis pigmentosa	*RP2*	c.358C>T c.371_378delAAGCTGGA	[43]
Retinitis pigmentosa	*PRPF31*	c.1115_1125del	[46]
Leber congenital amaurosis	*CEP290*	c.2991+1665A>G + c.2991+1665A>G	[18]
Leber congenital amaurosis	*CEP290*	c.2991+1665A>G + c.2991+1665A>G c.2991+1655A>G + c.5668G>T	[47,48]
Leber congenital amaurosis	*CEP290*	c.2991+1665A>G + c.2991+1665A>G	[49]
Leber congenital amaurosis	*CEP290*	c.2991+1665A>G + c.2991+1665A>G c.315del + c.316del	[50]
Leber congenital amaurosis	*LCA5*	c.835C>T + c.835C>T	[51]
Leber congenital amaurosis	*NPHP5*	c.421_422delTT + c.1036G>T	[24]
Non-syndromic retinitis pigmentosa	*USH2A*	c.8559-2A>G + c.9127_9129del	[52,53]
Non-syndromic retinitis pigmentosa/Usher syndrome (Type 2)	*USH2A*	c.2299delG + c.2276G>T c.2299delG + c.2299delG	[54]
Usher syndrome (Type 1)	*USH1B*	c.6070C>T + c.223G>C c.1996C>T + c.133-2A>G	[55]

## Data Availability

Not applicable.

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
