# Peer review of "Retinal Ciliopathies and Potential Gene Therapies: A Focus on Human iPSC-Derived Organoid Models"

_ijms, 2024, doi:10.3390/ijms25052887_

Round 1
Reviewer 1 Report
Comments and Suggestions for Authors
Overview: McDonald A and Wijnholds J provide a comprehensive, well-written, and organized review of human-induced pluripotent stem cell (iPSC)-derived retinal organoids for studying human retinal ciliopathies. The authors address an important problem that is of great interest to the reader. The authors provide substantial evidence to support the use of iPSC retinal organoids, while at the same time, discussing some of the disadvantages, using current literature sources. Overall, this review is informative and provides an in-depth current report of iPSC-derived retinal organoids and other technologies for the study of retinal ciliopathies.
Minor comments to the authors:
1. First sentence: Consider starting a new sentence after “cilium” and before “perturbations” to improve sentence structure.
2. In the Abstract, spell out all abbreviations at first mention.
3. Provide a statement for the rationale of this review and state the specific goal(s) of the review.
4. Provide a concluding paragraph to sum up the review, giving a take-home message.
Author Response
Dear reviewers,
Thank you for taking the time to read and asses our review article “Retinal Ciliopathies and Potential Gene Therapies: a Focus on Human iPSC-Derived Organoid Models”. We have taken your remarks into account and made adjustments to the manuscript. Hopefully we have sufficiently addressed your concerns.
Reviewer #1
- First sentence: Consider starting a new sentence after “cilium” and before “perturbations” to improve sentence structure.
Response: A new sentence has been created between “cilium” and “perturbations”.
- In the Abstract, spell out all abbreviations at first mention.
Response: L16 the abbreviation hiPSC has been spelled out as Human induced pluripotent stem cells.
- Provide a statement for the rationale of this review and state the specific goal(s) of the review.
Response: The aim of this review is to inform the reader of current retinal organoid models of retinal ciliopathies, along with their benefits and limitations. Additionally, to inform the reader of current gene therapy approaches, and developing technologies, targeting retinal ciliopathies. We believe this is sufficiently stated in the abstract L20-L24, and in the introduction L81-L83
- Provide a concluding paragraph to sum up the review, giving a take-home message.
Response: A new concluding remarks section has been added to the manuscript L700-L713
Reviewer 2 Report
Comments and Suggestions for Authors
Journal: International Journal of Molecular Sciences
Manuscript ID: ijms-2868226
Type of manuscript: Review
Title: Retinal Ciliopathies and Potential Gene Therapies: a Focus on Human
iPSC-Derived Organoid Models
Authors: Andrew McDonald, Jan Wijnholds *
Molecular Pathology, Diagnostics, and Therapeutics
This review article presents trials of gene supplementation and gene editing therapies for retinal ciliopathies-related genetic disorders. The authors describe the pathologies and diversity of ciliopathies-related genetic disorders and report on the latest developments in animal and organoid experiments. Overall, the review is careful and comprehensive and cites appropriate references.
Since the authors mentioned the diversity of cilia-related genetic disorders, this reviewer would have liked to see a more detailed description of what endpoints are verified in each (or representative) experimental system. For example, the authors describe the assessment of photo-transduction, irregular rod morphology, mislocation of opsins, the thickness of retina-related structures, and so on (L186-L197), but it is unclear whether those are feasible in organoids. Also, it should be clarified what phenotypic variety of diseases are (or not) represented by the organoid models. Furthermore, there is a description of the change in the amount of "marker" (around L368 and elsewhere), but without knowing how the marker relates to the progression or cause of the pathological changes, it is difficult to determine the effectiveness of the treatment. In short, this reviewer cannot see from this review whether the involvement of diverse cell types and morphological markers related to the eye can be evaluated with organoids. Therefore, this reviewer recommends that an explanation of this point be added in the introduction or the respective sections of the article. This reviewer feels that the technical description of gene therapy is sufficient.
Typo
L393 phenotype type -> phenotype
Author Response
Dear reviewers,
Thank you for taking the time to read and asses our review article “Retinal Ciliopathies and Potential Gene Therapies: a Focus on Human iPSC-Derived Organoid Models”. We have taken your remarks into account and made adjustments to the manuscript. Hopefully we have sufficiently addressed your concerns.
Reviewer #2
- “Since the authors mentioned the diversity of cilia-related genetic disorders, this reviewer would have liked to see a more detailed description of what endpoints are verified in each (or representative) experimental system. For example, the authors describe the assessment of photo-transduction, irregular rod morphology, mislocation of opsins, the thickness of retina-related structures, and so on (L186-L197), but it is unclear whether those are feasible in organoids.” “This reviewer cannot see from this review whether the involvement of diverse cell types and morphological markers related to the eye can be evaluated with organoids.
Response: We believe the reviewers primary concern is the lack of clarity on the validity of assessments that can be applied to retinal organoids. In regards to this a new paragraph has been added to section 2. iPSC-Derived Retinal Organoids as Models for Human Retinal Ciliopathies, L155-L183. Here we further elaborate on the techniques used to asses retinal organoid models, as well as referring to articles which discuss analytical techniques in greater detail.
- “Also, it should be clarified what phenotypic variety of diseases are (or not) represented by the organoid models”
Response: We believe sufficient information has been provided regarding the patient derived organoid models. Whether the patient from which the iPSCs were derived are classified as having retinitis pigmentosa, Leber congenital amaurosis, Usher syndrome type 1/2; along with the relevant genotype. This information can be found in the relevant sections or in table 1. A more detailed description of the patient phenotype is often not described in the referenced study.
- “Furthermore, there is a description of the change in the amount of "marker" (around L368 and elsewhere), but without knowing how the marker relates to the progression or cause of the pathological changes, it is difficult to determine the effectiveness of the treatment.”
Response: As elaborated further in the new text L155-L183, retinal cell types are known to express particular marker proteins which play a role in the function of the cell in question. Loss, reduction, or mislocalisation of these markers may indicate perturbations of cellular function. Indeed, this is not indicative of a definitive pathology, further analysis beyond a change in marker expression would be required to elucidate disease mechanisms. Instances in the manuscript where markers of interest were not specified by name have been adjusted, L334 and L396.
- Typo L393 phenotype type -> phenotype
Response: Typo previously on L393 (now L420) has been corrected.